Plant-insect interactions patterns in three European paleoforests of the late-Neogene—early-Quaternary

http://orcid.org/0000-0003-3693-9581 Adroit Benjamin 1 2 benjamin.adroit@umontpellier.fr
Girard Vincent 2
Kunzmann Lutz 3
Terral Jean-Frédéric 2
http://orcid.org/0000-0003-1592-0988 Wappler Torsten 1 4
1 Steinmann Institute for Geology, Mineralogy and Palaeontology, Division Palaeontology, Rheinische Friedrich-Wilhelms Universität Bonn , Bonn , Germany
2 Institut des Sciences de l’Evolution, UMR 5554, University of Montpellier , Montpellier , France
3 Senckenberg Natural History Collections Dresden , Dresden , Germany
4 Hessisches Landesmuseum Darmstadt , Darmstadt , Germany
De Baets Kenneth
Electronic publication date: 2018 Jun 20
Publication date: 2018
Volume: 6
Electronic Location ID: e5075
Received 2017 Nov 29; Accepted 2018 Jun 4
Copyright: © 2018 Adroit et al.
Copyright year: 2018
Copyright holder: Adroit et al.
License: This is an open access article distributed under the terms of the Creative Commons Attribution License, which permits unrestricted use, distribution, reproduction and adaptation in any medium and for any purpose provided that it is properly attributed. For attribution, the original author(s), title, publication source (PeerJ) and either DOI or URL of the article must be cited.
License URL: https://creativecommons.org/licenses/by/4.0/

Keywords: Herbivory, Berga, Seasonality, Bernasso, Willershausen Lagerstätte, Paleoclimate, Insect feeding, Plant-insect associations, Pliocene, Pleistocene

Funding: Deutsche Forschungsgemeinschaft (DFG) grant no WA 1492/8-1; 11-1 The present study was financially supported by grants of the Deutsche Forschungsgemeinschaft (DFG, grant no WA 1492/8-1; 11-1). The funders had no role in study design, data collection and analysis, decision to publish, or preparation of the manuscript.

==============================
Plants and insects are constantly interacting in complex ways through forest communities since hundreds of millions of years. Those interactions are often related to variations in the climate. Climate change, due to human activities, may have disturbed these relationships in modern ecosystems. Fossil leaf assemblages are thus good opportunities to survey responses of plant–insect interactions to climate variations over the time. The goal of this study is to discuss the possible causes of the differences of plant–insect interactions’ patterns in European paleoforests from the Neogene–Quaternary transition. This was accomplished through three fossil leaf assemblages: Willershausen, Berga (both from the late Neogene of Germany) and Bernasso (from the early Quaternary of France). In Willershausen it has been measured that half of the leaves presented insect interactions, 35% of the fossil leaves were impacted by insects in Bernasso and only 25% in Berga. The largest proportion of these interactions in Bernasso were categorized as specialist (mainly due to galling) while in Willershausen and Berga those ones were significantly more generalist. Contrary to previous studies, this study did not support the hypothesis that the mean annual precipitation and temperature were the main factors that impacted the different plant–insect interactions’ patterns. However, for the first time, our results tend to support that the hydric seasonality and the mean temperature of the coolest months could be potential factors influencing fossil plant–insect interactions.

Introduction

Climate is a major factor affecting the extension, structure and composition of terrestrial ecosystems (Taylor et al., 2012; Frank et al., 2015). Hence, past climatic oscillations are of special importance for understanding and interpreting biotic changes in the past (DeChaine & Martin, 2006) and are of interest in terms of forecasting the biotic response to future global warming (Meehl et al., 2007). Nowadays, it is clear that human activities have now reached a global impact affecting components of the Earth system as a whole (Turner et al., 1990; Heller & Zavaleta, 2009). In terrestrial ecosystems, arthropods are one of the most important components in biodiversity (Yang & Gratton, 2014) and their interactions with plants are essential for terrestrial food webs (Forister et al., 2015). Many modern ecological studies are focusing on these interactions between plants and insects but interpretations may be limited, therefore a combination with studies focused on the fossil record is necessary (Wilf, 2008). Studies on fossil insect herbivory have provided a variety of ecological and evolutionary information over long periods of time, such as climate (Wappler, 2010; Wappler et al., 2012), the evolutionary impact of plant radiations (Labandeira, 2012; Labandeira & Currano, 2013), food web dynamics (Wappler & Grímsson, 2016), extinction patterns (Labandeira, 2002; Labandeira, Johnson & Wilf, 2002; Donovan et al., 2016), and ecosystem recovery after extinction events (Wappler et al., 2009; Labandeira, Kustatscher & Wappler, 2016). They have also shown that biodiversity loss may greatly impede trophic interactions and change the overall food web structure of ecological systems (Haddad et al., 2009). Moreover, there is increasing concern about the loss of biological diversity from ecosystems (Hooper et al., 2012).

The large amount of Plio–Pleistocene fossil records offers an exceptional possibility to estimate the evolution and dynamics of associations between plant species and their dependent insect-herbivore species, as descriptions of Plio–Pleistocene floral changes by combining different and complementary data (Tzedakis, Hooghiemstra & Pälike, 2006; Médail & Diadema, 2009; Postigo Mijarra et al., 2009; Magri, 2010; Migliore et al., 2012). Although a few isolated records of specialized phytophagy categories have been reported from the Pliocene (Straus, 1977; Givulescu, 1984; Titchener, 1999), only a single systematic survey of plant–arthropod interactions has been carried out on an early Pleistocene flora (Adroit et al., 2016).

Thus, an ideal setting for the evaluations of relationships among global climate and biodiversity under conditions warmer than today, but with a similar paleogeographic configuration (Raymo et al., 2011; Rohling et al., 2009) is possible throughout the famous upper Pliocene fossil Lagerstätten Willershausen (3.2–2.6 Ma; MN 16/17) (Hilgen, 1991; Mai, 1995) and the comparisons with Berga (Germany, late Pliocene) and the French Pleistocene locality of Bernasso (Adroit et al., 2016). Willershausen and Berga outcrops are of similar age (Piacenzian) and are located in the surroundings of the Harz Mountains, Germany (Fig. 1). The Willershausen paleoforest was dominated by typical taxa of hilly mesophytic woodland (Ferguson & Knobloch, 1998; Knobloch, 1998) such as Acer, Aesculus, Carpinus, Fagus, Quercus, Sassafras, Tilia (Mai, 1995; Knobloch, 1998) and other taxa such as Parrotia, Zelkova and Liquidambar were also characteristic elements of Willershausen (Mai, 1995). All of these taxa were also found in Berga (Mai & Walther, 1988). The presence of these taxa indicates relatively warmer conditions in Europe than today during the late Pliocene (Uhl et al., 2007; Thiel, Klotz & Uhl, 2012). Most plant fossil evidence from Central Europe outcrops (Haywood, Sellwood & Valdes, 2000; Uhl et al., 2007; Williams et al., 2009; Thiel, Klotz & Uhl, 2012) data from marine isotopes, and geological evidence (Driscoll & Haug, 1998; Haug, Tiedemann & Keigwin, 2004) also support the warmer climate condition estimated for the late Pliocene. Bernasso is younger than the German outcrops, estimated at around 2.16–1.96 Ma (Suc, 1978; Leroy & Roiron, 1996). It is located in southern France, 5 km far away from Lunas in the department of Hérault (Suc, 1978; Leroy & Roiron, 1996; Adroit et al., 2016). The Bernasso fossil leaf assemblages are mainly dominated by the genera Carpinus, Parrotia, Acer and Sorbus (Leroy & Roiron, 1996; Adroit et al., 2016) wherein many plant species are in common with the German fossil leaf assemblages. Detailed descriptions are available in Leroy & Roiron (1996) and Adroit et al. (2016). The decreasing temperatures, from ca. 18 to 14 °C throughout the Pliocene (Thunell, 1979; Ravelo et al., 2004; Hansen et al., 2013) lead to the dominant European vegetation changing gradually from highly diverse subtropical and warm-temperate forests to temperate deciduous forests with East Asian and partly North American affinities (Mai, 1995).

Figure 1 The location of Willershausen and Berga outcrops (Germany) from the late Pliocene.

(A) The location of Germany in Europe. (B) The locations of both outcrops in Germany. (C) Zoom on the area near Göttingen. On the scale, each dash (black or white) represents 5 km. The data from the maps in (A) and (B) come from Natural Earth database (http://www.naturalearthdata.com).

Through the comparison of three European forest plant communities of the Plio–Pleistocene, the aim of this study was to understand how climatic parameters could have impacted plant–insect interactions of fossil leaves. It has been expected that the difference of estimated mean annual temperatures (MATs) between those paleoforests could have a major impact on the quantity of the plant–insect interactions (Coley & Aide, 1991; Zvereva & Kozlov, 2006; Currano, Labandeira & Wilf, 2010). Moreover, the estimated mean annual precipitations (MAPs) of the fossil outcrops should be negatively related to the gall proportions observed on the fossil leaves (Fernandes & Martins, 1985; Fernandes & Price, 1988; Price et al., 1998; Lara, Fernandes & Gonçalves-Alvim, 2002). Our results provide the first approach on plant–insect interactions from the Plio–Pleistocene in European paleoecosystems.

Study Area

Willershausen, Lower-Saxony, Germany

Geological studies of Willershausen date back to the end of the 19th century (Wegele, 1914) (see details in Wegele, 1914; Ferguson & Knobloch, 1998; Meischner, 2000). The absence of bioturbation gave rise to one of the exceptionally well preserved floras and faunas (Briggs et al., 1998). The Willershausen site was a lake which developed in a pond due to the dissolution of underlying Permian evaporites and has been buried by Triassic and Early Jurassic sediments (Briggs et al., 1998; Meischner, 2000; Kolibáč et al., 2016). Today, Willershausen is an abandoned clay mining operation and it is included in the Geopark Harz, Braunschweiger Land, Ostfalen since 2012. This paleolake was ca. 200 m in diameter and approximately 10 m deep with a narrow sand beyond which the sides inclined abruptly toward the bottom of the lake (Meischner, 2000). Willershausen geology has been described by Von Koenen, 1895 and detailed compilations can be found in Vinken (1967), Ferguson & Knobloch (1998) and Meischner (2000).

The leaves used in this study are stored in different museum collections in Germany. The majority (6,546 leaves) is located at the Geoscience center of the University of Göttingen (GZG.W collection). Additional fossil leaves are stored in the Staatliches Museum für Naturkunde Stuttgart (SMNS.W collection; 957 leaves), in the collections of TU Clausthal of Clausthal-Zellerfeld (320 leaves), in the Naturkundemuseum im Ottoneum of Kassel (NMOK.W; 236 leaves) and in the Senckenberg Natural History Collections Dresden (14 leaves). Some of the best well-preserved fossil specimens are presented in Fig. 2A. The flora from Willershausen comprised a rich vegetation community including the presence of Acer, Alnus, Betula, Carpinus, Carya, Fagus, Pterocarya, Populus, Quercus, Tilia, Ulmus, Zelkova (Straus, 1977; Ferguson & Knobloch, 1998; Knobloch, 1998). The vertebrates Anancus (Mastodon) arvernensis and Tapirus were found in Willershausen and seems to indicated a Piacenzian age (late Pliocene, ca. 3.2–2.4 Ma; MN 16/17; (Mai, 1995), which is corroborated by the presence of Parrotia persica and Liquidambar europaeum (Mai, 1995).

Figure 2 Well-preserved samples of fossil leaves morphotypes from Willershausen and Berga outcrops, late Pliocene from Germany.

Plate 1. Fossil leaves from Willershausen (Göttingen coll.). (A) Ulmus carpinifolia with Hole feeding (DT05). (B) Alnus spaethii with Margin feeding (DT14). (C) Fagus sp with Piercing & Sucking (DT168); enlarged in (D). (F) Ulmus campestris with Mining (DT109) enlarged in (E). (G) Quercus praeerucifolia with Galling (DT145); enlarged in (H). (J) Ulmus carpinifolia with Skeletonization (DT17). (K) Populus tremula with Surface feeding (DT30); enlarged in (I). Plate 2. Fossil leaves from Berga (Dresden coll.). (L) Cercidiphyllum crenatum. (M) Fagus attenuata with Hole feeding (DT01). (N) Juglans sp. with Galls (DT34). (O) Pterocarya paradisiaca. (P) Quercus pseudocastanea with Galling (DT116). (Q) Quercus castaneifolia. White scale bar represents 1 cm; black scale bar represents 0.5 cm. Photographs by Benjamin Adroit.

Most plant fossil evidence from Willershausen indicates warmer conditions than today (Ferguson & Knobloch, 1998). The MAT in Willershausen was estimated between 10.6 and 15.6 °C on the base of the leaf morphology and of diversity of plant species niches (Table S1; Uhl et al., 2007); these different approaches explain the wide range of temperature estimated. The mean temperature of the coldest month (CMMT) is estimated between 0.6 and 3.2 °C and the MAP between 897 and 1151 mm per year (Table S1; Uhl et al., 2007; Thiel, Klotz & Uhl, 2012).

Berga, Thüringen, Germany

Berga was a lake in which compressions and impressions (some with the cuticles preserved) of leaves were found in silty sediments (Mai & Walther, 1988). It is 70 km far from the Willershausen outcrop. The stratigraphic age of the Berga sediments is estimated on the basis of sedimentological correlations referring to the Piacenzian (ca. 3–2.6 Ma) (Bachmann et al., 2008).

This leaf collection (534 specimens) is housed in the collection of the Senckenberg Natural History Collections Dresden, Germany. It contained many fossils of different origins (Mai & Walther, 1988), including 30 angiosperms leaf taxa (Fig. 2B). They represent different environments: a freshwater plant community, a swamp and riparian associations and a zonal mixed broadleaved conifer forest (which dominates the taphocoenosis). The temperatures were estimated with the same approach as Willershausen; MAT is estimated between 7.4 and 16.6 °C, the CMMT is between −4.3 and +0.6 °C and the MAP is between 897 and 1,297 mm per year (Table S1; Uhl et al., 2007; Thiel, Klotz & Uhl, 2012).

Bernasso, France

Bernasso was a lake developed when a basaltic flows shut off a canyon valley (Leroy & Roiron, 1996). Diatomites were formed and fossil leaves, often with rest of cuticle, were preserved. It is located close to Lunas (Hérault, Southern France) (Leroy & Roiron, 1996; Adroit et al., 2016). The fossil deposit is dated from the early Pleistocene on the basis of K/Ar analysis (Ildefonse et al., 1972) on a basaltic dyke that crosses the diatomite layers. A complementary analysis on cyclostratigraphy (Suc & Popescu, 2005) and paleomagnetism (Ambert et al., 1990) corroborated these results and estimated an age around 2.16–1.96 Ma.

The collection included 800 fossil leaves and 535 specimens well-preserved were described in (Adroit et al., 2016). These same specimens were also used for comparisons in the present study. The fossil leaves were conserved the Institut des Sciences de l’Evolution de Montpellier in France. Different preparation of fossil leaves were done by (Leroy & Roiron, 1996) and their impact on interpretation were discussed in (Adroit et al., 2016). The flora represents a mesothermic forest, mixing Mediterranean and Caspian elements (Suc, 1978; Leroy & Roiron, 1996). The MAT in Bernasso was estimated about 14–15 °C and the MAP is around 1,500 mm/year (Table S1; Leroy & Roiron, 1996). It is important to note that CLAMP results in Bernasso suggest a possibly lower temperature around 7 °C (Table S1; V. Girard et al., 2017, unpublished data).

Data Analyses

Plant–insect interaction identifications

The plant–insect interactions were identified following the “Guide to Insect (and Other) damage types on compressed plant fossils” (Labandeira et al., 2007). The damages type (DT) are easily recognizable thanks to the black reaction mark surrounding them (Labandeira, 2002; Labandeira et al., 2007). They are divided in seven functional feeding groups (FFG): hole feeding, margin feeding, skeletonization, surface feeding, piercing & sucking, mining and galling. Leaves without damage were also categorized in an eighth FFG called the undamaged leaves. The leaves undamaged has been take into account as a proxy of the non-palatability of the leaves, thus can be considered as another FFG. The leaves were examined under a binocular Leica MZ95 and all photographed with a Canon EOS 350D camera fitted with a Canon EF-S 60 mm f/2.8 macro lens. A Nikon Coolpix E4500 was used sometimes for precise pictures through the binocular. All pictures were developed using Adobe Lightroom CC v.2015 especially in order to improve contrast of the leaf. The insect interactions on leaves were scored according to the richness, frequency and distribution on the different plant species for each outcrop. For each DT, a host specificity value has been attributed by Labandeira et al. (2007) that allowed to classify our DTs into generalist interactions (made by polyphagous organisms) and specialized interactions (made by monophagous organisms) (Labandeira, 2002). Detailed plates of fossil leaves from Willershausen are available in Table S2 including the original descriptions of the plant–insect interactions made by (Straus, 1977) and our actual updates with the guide of insect (and other) damage types on compressed plant fossils (Labandeira et al., 2007).

The results obtained for Willershausen and Berga were compared to those recently published for the outcrop of Bernasso (Adroit et al., 2016). For some comparisons with Bernasso, new values were calculated based on raw data.

Statistical analyses

For each outcrop, the statistical analyses were performed on two different databases as described in Knor et al. (2012). The first one is the whole assemblage of plant–insect interactions. The second one considers only the interactions of the species that are significantly represented (more than 20 leaves). The quantitative analyses were done in R version 3.1.2 (R Development Core Team, 2014). The differences among the proportions of occurrences from all FFG were tested with Chi-squared-test. The remaining information needed for this test was obtained by using the generalized linear model of binominal distribution. Sample-based rarefaction curves were done to compare the different damage richness and the different plant richness between the outcrops (Gotelli & Colwell, 2001). At last, in order to observe the distributions of plant species according to the FFG among the different fossil leaf assemblages, principal component analysis (PCA) were performed with the software Past3 (v3.14) (Hammer, Harper & Ryan, 2001) in a biplot. PCA has been useful to assess relationships of the plant species for each FFG. The data matrices used for it considered the frequency of each eight FFGs for each plant species of each outcrop (i.e., for each outcrops a matrix such as FFG frequency × leaf morphotype).

Results

Comparisons of insect interactions and plant species richness

In Willershausen 50.4% of the leaves are damaged, and only 25.1% in Berga. This percentage was 34.6% in Bernasso (Adroit et al., 2016). These differences are statistically significant (p < 0.001) (Fig. 3; Table 1).

Figure 3 Quantitative distribution of plant–insect interactions from Willershausen, Berga outcrops (late Pliocene) and the fossil deposit of Bernasso (early Pleistocene).

Bernasso data come from the publication of Adroit et al. (2016). For each damage frequency, significant difference (α < 0.05) from an outcrop to another one is marked by an asterisk. The percentage of generalized and specialized damages are computed only with the damaged leaves; consequently their sum on each outcrop is 100% in this figure. According to the whole amount of leaves the percentage of generalist interactions are 42.8% for Willershausen, 17.8% for Berga and 19.8% for Bernasso and the percentage of specialist interactions are 11.2% for Willershausen, 8.4% for Berga and 17.9% for Bernasso.

Table 1 Frequency of the leaves damaged per FFG based on the whole flora.

Outcrops	# of leaves	Damaged	Generalist	Specialist	External_Specialized	Galling	Mining	MarginF	HoleF	Skeletonization	SurfaceF	P&S	
WILLERSHAUSEN	7,932	50.43	42.80	10.16	1.11	7.01	1.59	9.86	26.94	11.01	1.64	1.10	
BERGA	534	25.09	17.79	7.12	1.31	6.18	0.19	1.87	12.73	2.62	2.06	0.94	
BERNASSO	535	34.58	19.81	15.70	2.24	11.78	1.68	7.10	9.72	7.66	0.93	0.00	
Notes:

It happens that there is more than one FFG on a leaf damaged; consequently the sum of the percent of galling, mining, margin feeding, hole feeding, skeletonization, surface feeding and piercing & sucking exceed the value of the damaged leaves. Bernasso data originate from Adroit et al. (2016). Details of damaged leaves per species are presented in Table S3.

The frequencies of generalist interactions are 42.8% for Willershausen, 17.8% for Berga and 19.8% for Bernasso (Adroit et al., 2016). Only Willershausen frequency is significantly different from the others (p < 0.001). Willershausen leaves have especially much more hole feedings (26.9%) and margin feedings (9.9%) than Berga (respectively 12.7% and 1.9%) and Bernasso leaves (respectively 9.8% and 7%) (Fig. 3).

The frequencies of specialized interactions are 11.2% for Willershausen, 8.4% for Berga and 17.9% for Bernasso (Adroit et al., 2016). Only the Bernasso frequency is significantly different from the others (p < 0.001). This difference is mainly due to the important quantity of galling in Bernasso (12%) which is significantly higher than in Willershausen and Berga, respectively 7% and 6% (p < 0.01) (Fig. 3).

Rarefaction tests on plant species richness highlight that Willershausen has more plant species (>100) than Berga (33) and Bernasso (20) which has the less one (Fig. 4). However, the DT richness in Willershausen (36 DTs) and Berga (25 DTs) are lower than in Bernasso (40 DTs) (Fig. 4).

Figure 4 Rarefaction curves on the leaves from Willershausen, Berga (Germany; late Pliocene) and Bernasso (France, early Pleistocene).

Bernasso data come from the publication of Adroit et al. (2016). The gray curves represent Berga, the blue curves represent Willershausen and the orange curves represent Bernasso. The shaded area represents the standard deviation below and above the average of the resamples, with the method from Heck, van Belle & Simberloff (1975). Rarefaction curves represent the number of specimens by: (A) richness of plant species; (B) richness of damage type (DT); (C) richness of generalized damage; (D) richness of specialized damage.

Structure of the paleoforests with the damage distribution on plant species

Figure 5 presents the different PCA realized for the three outcrops with the data of plant and DT diversities. For each outcrop, only the first two axes are presented as for Willershausen they represent 77% (Fig. 5A), for Berga 93% (Fig. 5B) and for Bernasso 91% (Fig. 5C) of the whole distribution.

Figure 5 Principal components analysis (PCA) based on the proportion (in percentage) of the FFG on each plant species.

(A) Willershausen (Germany, late Pliocene), (B) Berga (Germany, late Pliocene), (C) Bernasso (France, early Pleistocene). Circles are used to highlight the common plant species between the different outcrops. Due to large amount of plant species names on (A) Willershausen, the species names near the axes intersection were replace by alphabetic letter for visibility concerns: a, Malus pulcherrima; b, Prunus mahaleb; c, Fagus sylvatica; d, Sorbus gabbrensis; e, Populus tremula; f, Populus willershausensis; g, Zelkova carpinifolia; h, Betula pubescens; i, Betula_sp1; j, cf Toona.

For Willershausen (Fig. 5A) the FFGs hole feeding and skeletonization are positively correlated with PCA-axis 1 (respectively, 0.76 and 0.61) and undamaged is negatively correlated with this axis (−0.97) (Data S1). Skeletonization and galling are positively correlated with PCA-axis 2 (respectively, 0.62 and 0.63) while hole feeding is negatively correlated with this axis (−0.73) (Data S1). Concerning the species, three pools of plant species can be distinguished. The Tilia (T. saportae, T. cf. saviana), the Ulmus (U. carpinifolia, U. campestris), the Fagus (F. grandifolia, F. pliocenica), Acer integerrimum and Quercus roburoides are all along the positive part of the PCA-axis 1. The leaves of these species have the highest DT frequency of hole feeding and skeletonization. A second set of taxa is composed, for the most evident species, by Acer cappadocicum, Acer laetum, Carya minor, cf. Magnolia sp1 and 2, Populus willershausensis, Q. praeerucifolia and Zelkova ungeri. They are along the negative part of the PCA-axis 1 and along the positive part of the PCA-axis 2. They are mainly affected by the FFG galling (specialized interaction) or have no damage. At last, the third set of species is composed of Fagales (F. sylvatica, all the Quercus, Alnus and Betula species) and is in the negative part of the PCA-axis 2. These leaves are mainly undamaged or only impacted by hole feeding (generalist interaction).

For Berga (Fig. 5B) the FFGs hole feeding, skeletonization and undamaged are positively associated with the PCA-axis 1, respectively with a correlation of 0.56, 0.73 and 0.99 (Data S1). Hole feeding and skeletonization are also correlated with the PCA-axis 2, negatively for hole feeding (−0.79) and positively for skeletonization (0.66) (Data S1). Concerning the species, we can note that Taxodium dubium, Z. ungeri, Cercidiphyllum crenatum and A. integerrimum are correlated with this undamaged category. F. attenuata, Acer tricuspidatum and Quercus sp. are mainly correlated with hole feeding.

For Bernasso (Fig. 5C; Adroit et al., 2016), the skeletonization and galling are positively correlated with the PCA-axis 1 (0.82 and 0.92) while undamaged is negatively correlated with this axis (−0.94) (Data S1). Hole feeding and skeletonization are positively correlated with PCA-axis 2 (0.92 and 0.25) while undamaged and galling are negatively correlated with this axis (−0.25 and −0.37) (Data S1). Acer monspessulanum and Sorbus domestica are in the positive part of the PCA-Axis 1 while the other are in the negative one (to note that Parrotia persica is close to zero). Concerning the PCA-axis 2, A. monspessulanum, Carpinus orientalis and Carya minor are in the positive part of the PCA-axis 2 while the others are in the negative part (to note that Z. ungeri is close to zero).

Furthermore, Z. ungeri is a species found in the three outcrops (Fig. 5) and compared to its position in the different PCAs, we can note that Z. ungeri is mostly associated with the FFG undamaged. However, for other common plant species, their relative position on the PCAs could be different. A. integerrimum in Berga (Fig. 5B) is mostly associated with the FFG undamaged while in Willershausen is opposite to this FFG as it is mainly associated to the FFG hole feeding and more weekly with skeletonization, margin and galling (Fig. 5A).

Comparing Willershausen and Bernasso (Figs. 5A and 5C), A. monspessulanum is mainly associated to skeletonization and galling in Willershausen (Fig. 5A) and in Bernasso it is with skeletonization, galling too but also with hole feeding (Fig. 5C).

Sorbus domestica and Carpinus orientalis are both associated to the FFG undamaged in Willershausen (Fig. 5A). In Bernasso, S. domestica is associated with galling and skeletonization and C. orientalis is associated with hole feeding and undamaged (Fig. 5C). Carya minor is associated to skeletonization and galling in Willershausen (Fig. 5A) while in Bernasso it is associated to undamaged and hole feeding (Fig. 5C). Parrotia persica is associated to galling and undamaged in Bernasso (Fig. 5C) while in Willershausen, it is associated with the FFGs skeletonization and galling (Fig. 5A).

Discussion

Floristic richness and herbivory representativeness

All genera and at least 22 plant species from Berga leaf assemblage are also present in the Willershausen assemblage (Table S3). It can be explained by the geographical and stratigraphical proximity of the two outcrops. Bernasso had nearly the same composition of plant genera found in Willershausen (except Ilex only found in Bernasso) and also the majority of plant species (Table S3) despite its geographical situation and its younger age. There is quite a difference of plant richness between Bernasso and Berga, but the genera are the same (Table S3). This may suggest a difference in specific richness between those paleoforests. Rarefaction data indicated for the Willershausen leaf assemblage a highest plant species richness than the ones of Berga and Bernasso (Fig. 4). However, the original sample size is considerably larger in Willershausen and could have led to artificial differences of plant species richness between the outcrops (Table 1). However, a bias due to the sample size is unlikely as Bernasso has the highest DT richness while the plant species and the quantity of leaves are lower than the one of the Willershausen assemblage.

Sampling effort tests indicate that enough specimens were taken into account to have a representative overview of the interactions on plant species found into the different outcrops (Fig. 4). The large standard deviation observable on the Willershausen rarefaction curves on Fig. 4 is due to this size of the fossil collection that includes around 8,000 specimens while the others are only 534 for Berga and 535 for Bernasso.

Relations between herbivory and the different mean annual temperatures estimated

Climatic conditions seem to be in relation with variations in richness and frequency of plant–insect interactions (Currano, Labandeira & Wilf, 2010). If an increase of temperature seems to stimulate insect herbivory (Coley & Aide, 1991; Coley & Barone, 1996; Zvereva & Kozlov, 2006; Currano, Labandeira & Wilf, 2010), it is still difficult to understand the complete role of temperature in the modulation of herbivory (DeLucia et al., 2012).

Thiel, Klotz & Uhl (2012) indicated, through leaf morphological analyses, that temperatures estimated for Willershausen were approximately 3 °C higher than those for Berga. These paleoforests were geographically very close to each other (less than 70 km) and at a similar latitude (51°N) (Fig. 1). Today the nearest meteorological stations of these locations (Willershausen: Göttingen, Lower-Saxony; Berga: Nordhausen, Thüringen) indicates the same MAT also for the coldest and warmest months over the last years (http://www.worldweatheronline.com). Such current similarities make the argument for similar paleoclimates of the two fossil localities if they were strictly of the same age. However, between 3 and 2.5 Ma, CO2 concentration progressively decreased (Kürschner et al., 1996; van de Wal et al., 2011) implicating a continuous decrease of MATs (Willeit et al., 2015). Consequently, as Willershausen was warmer than Berga, the paleoforest of Willershausen grew under higher atmospheric CO2 concentration than the Berga paleoforest. It seems to corroborate by the higher damage frequency observed in Willershausen that can have been favored by an increase of C/N ratios and an increase of photosynthesis rates (due to the high CO2 concentration) (Bezemer & Jones, 1998; Stiling & Cornelissen, 2007; DeLucia et al., 2012). However, Willershausen and Berga had different sedimentological contexts and the preservation of the fossil leaves did not follow the same taphonomical constrains in the two outcrops. This could have influenced interpretation of the climate through morphological analyses.

For this reason, Thiel, Klotz & Uhl (2012) were in favor of the Coexistence Approach for climate interpretation which estimated similar temperature for Berga and Willershausen. It has been highlighted that the diversity of insects is often correlated to richness of plant species (Siemann, Tilman & Haarstad, 1996; Wright & Samways, 1998; Knops et al., 1999; Mulder et al., 1999) and should be expected to have higher damage richness in the more diverse paleoforest (Price, 1991, 2002). Thus, the higher richness and frequency of damage in Willershausen than in Berga could also be due to a higher insect diversity. Nevertheless, despite its higher plant richness Willershausen had less DT richness than the fossil leaf assemblage of Bernasso (Figs. 3 and 4). Bernasso had also more damage richness and frequency than Berga (Figs. 3 and 4). Thus, these observations make this assumption unsustainable for our study. It is also conceivable that the relative abundance of a plant species in those paleoforests could partly explain the herbivory measured. Indeed, more a plant species is represented in the forest community, then more individuals of the plant species have had a chance to be damaged by insect feeding (Feeny, 1976). Unfortunately, for the fossil record, it is not possible to support this assumption because the leaf quantity of a plant species from an outcrop cannot be correlated to the relative abundance of this plant species in the paleoforest. Consequently, the difference of plant species richness observed between the outcrops could be firstly due to differences of fossil preservation than more to differences among paleoecosystems.

For Bernasso, the latitudinal position is different from Berga and Willershausen, as it located 1,000 km to the South. It has been highlighted that the insect diversity increases getting closer to the tropics (Hutchinson, 1959; Klopfer, 1959; Klopfer & MacArthur, 1960; MacArthur, 1972; Coley & Barone, 1996; Fraser, 2017). The southern position of Bernasso could partly explain the measured damage type richness. Nevertheless, the quantity of damage is not exclusively linked to the insect diversity (Currano, Labandeira & Wilf, 2010). Latitudinal differences could led to a difference of thermal seasonality (Saikkonen et al., 2012) which is the key to the latitudinal gradient of insect diversity (Archibald et al., 2010). Leroy & Roiron (1996) indicated that Bernasso paleoforest grew under temperatures of 14–15 °C and precipitations around 1,500 mm/a. Recently, V. Girard et al. (2017, unpublished data) re-estimated Bernasso climate with different approaches and some results, based on leaf morphological traits, estimated temperatures in Bernasso to be cooler than estimations of Leroy & Roiron (1996), while the pollen analysis from the same study tend to corroborate previous estimations done by Leroy & Roiron (1996).

Relations between herbivory rates and temperatures of the coldest months

Berga has a low temperature of the coldest months (from −6.4 to 2 °C) compared to Willershausen which had the highest temperatures (from −0.5 to 5.1 °C) (Uhl et al., 2007; Thiel, Klotz & Uhl, 2012). These lower temperatures during the cold period could explain the lowest damage frequency observed in Berga. Indeed, insects are poikilotherms, meaning that their body temperature is extremely dependent to the environment temperature (Meglitsch, 1972). Cooler temperatures decrease the insect metabolism (leading to diapause of insects) and the quantity of generations per year (Archibald et al., 2010), consequently it could also reduce the herbivory rates during the year (Bale & Hayward, 2010). Concerning Bernasso, the different estimations of temperatures, included the CMMT, are lower than those of Willershausen (Uhl et al., 2007; Thiel, Klotz & Uhl, 2012), thus the lowest frequency of damage could also be due to a lower insect metabolism in Bernasso than in Willershausen. The lowest frequency of damage in Berga than in Bernasso could also be due to insect diapause in the case of coolest temperatures being lower in Berga. However, the estimated temperatures of Bernasso overlap with the ones of Berga (especially for the coolest temperatures) and therefore complicate any interpretations about the damage frequency between these two fossil leaf assemblages.

Moreover, it is important to note that no data about insect richness of these different paleoforests are available. Although it could be assumed that insect richness between Willershausen and Berga could be similar because outcrops are geographically and temporally similar, the insect richness of Bernasso could be quite different. Consequently, in cases of differences in the insect faunas, the previous relation could be disturbed as some insects, such as larvae of Thaumetopoea pityocampa, feed on plants during the winter season (Battisti et al., 2005; Buffo et al., 2007), when others insects have no or lower activity (Hahn & Denlinger, 2007).

More precision provided by proportion of generalized/specialized damages

The comparison of plant–insect interaction between different locations or through different time periods could still be upset by local disturbances (fires, floodings, etc.) or other constraints (such as different soils) that are not perceptible in fossil record and could impacted damage pattern in general (Currano et al., 2011; García, Castellanos & Pausas, 2016). Moreover, taphonomic biases, especially fossil preservation and different excavation histories, could also interfere with our analyses. For example, the damage frequency observed in fossil record could be partly distorted because the damaged leaves had less chance to be preserved in the fossil record than the complete and undamaged leaves (Ferguson, 2005). For all these reasons, we suggested complementing analyses by comparison of the proportions of generalized and specialized damage patterns.

Leckey et al. (2014) indicate that the proportion of generalist and specialist herbivores may change between different forests because the difference of abiotic parameters (such as climate). There are the lowest proportions of specialist interactions (mainly based on galling) in Willershausen and Berga, and conversely the highest proportion is in Bernasso (Fig. 3), this may due to climatic factors (Leckey et al., 2014). Indeed, precipitation in Bernasso was higher than in Willershausen and Berga (Leroy & Roiron, 1996; Uhl et al., 2007). Also, the hydric seasonality was possibly more important in Bernasso as indicated by some CLAMP estimations (other CLAMP estimations minimize Bernasso hydric seasonality; V. Girard et al., 2017, unpublished data. This is also in agreement with the supposed Mediterranean climate for Bernasso (based on the plant species diversity; Leroy & Roiron, 1996) that provided heavy constrain to plants here due to less water availability during the dry season (Bagnouls & Gaussen, 1957; Daget, 1977, 1984). The higher seasonality conditions in Bernasso compared to conditions proposed for Berga and Willershausen could also be supported the idea that regional conditions of Northern Atlantic realm were more marked by higher seasonality during Pleistocene than the Pliocene (Williams et al., 2009; Hennissen et al., 2015; Utescher et al., 2017). Water stress should have a positive impact on galling quantity, as many studies already mentioned that galling is an adaptation of stressful environment (Fernandes & Martins, 1985; Fernandes & Price, 1988, 1992; Price et al., 1998; Lara, Fernandes & Gonçalves-Alvim, 2002). In addition, Cuevas-Reyes et al. (2003), who studied the development of galling, showed that it exists a negative correlation between gall-forming insect species richness and plant species richness. It could also partly explain the highest proportion of specialized interactions in Bernasso. Additionally, a forest in its late successional stage, as it has been proposed for Bernasso (Leroy & Roiron, 1996; Adroit et al., 2016), tend to favor the richness of gall-inducing insects (that increase the proportion of specialized interaction) (Fernandes, Almada & Carneiro, 2010; Adroit et al., 2016).

Inputs of the comparisons between the common plant species from the different outcrops

This global comparison of specialized and generalized damages between the fossil leaf assemblage of Bernasso, Willershausen and Berga are also observable precisely on the common plant species statistically represented in each outcrop. However, the FFGs and especially the undamaged feature on some plant taxa are similar or could be slightly different between the fossil leaf assemblage (Fig. 5). It tends to confirm that the abiotic parameters are important determinant factors involving significant variation of herbivory between different paleoenvironments (Cuevas-Reyes et al., 2004, 2003; Leckey et al., 2014). Biotic parameters can also be involved in the difference of interaction structures. For example, a decrease in food quality caused by higher concentration of carbon in plants could also have a negative impact on herbivory (Stiling & Cornelissen, 2007), but in general, it is compensated by an increase of insect feeding (Bezemer & Jones, 1998). The impact of biotic factors seems to be further confirmed as in Willershausen we can note that most Fagales (Betulaceae: Alnus, Betulus, Carpinus; Fagaceae: Fagus, Quercus; Juglandaceae: Carya, Juglans) are all associated to hole feeding and to undamaged feature (Fig. 5A). This measurement cannot be due to hazard but it probably reflects an effect of some biotic parameters (such as genetic background, plant competition, host specificity, etc.).

Conclusion

Despite their similar plant species and their relative geographical and stratigraphical proximity (at least for Berga and Willershausen), trophic structures of those paleoforests were different. On the contrary to different hypotheses made on previous studies (Fernandes & Martins, 1985; Fernandes & Price, 1988; Coley & Aide, 1991; Coley & Barone, 1996; Price et al., 1998; Lara, Fernandes & Gonçalves-Alvim, 2002; Zvereva & Kozlov, 2006; Currano, Labandeira & Wilf, 2010), there was no relationships between the MAT and the quantity of plant–insect interactions, as well as between the MAP and the proportion of some specialist damages. The comparison of the fossil records of Willershausen, Berga and Bernasso allowed discussion about the potential impacts of seasonality of the precipitation on the high proportion of galling. In addition, results suggested also that the herbivory rates could be impacted by the CMMTs of these paleoenvironments. Such observations can be related to the insects’ response to climatic variation, which is very sensitive (Bale & Hayward, 2010). The next step should be to conduct a meta-analysis in order to improve the knowledge of the relations between plant–insect interactions and climate, to this end, further studies are needed. Concerning European paleoforests, the studies of other late Pliocene outcrops such as the one of Frankfurt-am-Main in Germany (Thiel, Klotz & Uhl, 2012) or Fossano in Italy (Macaluso et al., 2018) should provide interesting data in that way. Lastly, this study points out that comparisons of plant–insect interactions from different paleoforests are limiting by the fossil preservation which significantly affects the fossil leaf collections available in the outcrops. Comparisons with some similar modern forests could be relevant in order to better discern the differences in proportions of plant species in the paleoforests communities.

Supplemental Information

Supplemental Information 1 Table S1. Estimation of the mean annual temperature (MAT), mean temperature of the coldest month (CMMT) and the mean annual precipitation (MAP).

These are based on the following methods: co-existing approach (CoA), leaf margin analysis (LMA), Climate Leaf Analysis Multivariate Program (CLAMP), European Leaf Physiognomic Approach (ELPA), Climatic Amplitude Method (CAM). For further details, please note that data for Willershausen and Berga come from Thiel et al. (2012) and data for Bernasso come from a- Leroy & Roiron (1996), b- Girard et al. (submitted).

Click here for additional data file.

Supplemental Information 2 Table S2. Plant-insect interactions and herbivory patterns observed and originally described by Adolf Straus (1977) in his first comprehensive work on leaf interactions in Willershausen (Germany, late Pliocene).

¶ not found in the collections; * new ichnospecies type; § named after recent species with the addendum fossilis; H: holotype; P: paratype.

Click here for additional data file.

Supplemental Information 3 Table S3. General spreadsheet of plant species considered statistically (more than 20 specimens) for each outcrop.

Results are expressed in percent. The plant species with less than 20 specimens were taken into account for the plant species richness and the general result of FFGs in each fossil leaf assemblages.

In Willershausen (Germany, late Pliocene) the non-statistical species are: Populus monilifera, Acer opalus, Fagus pliocenica, Fraxinus ornu, Carya serrifolia, Prunus avium, Sassafras ferretianum, Juglandaceae sp1, Sorbus praetorminalis, Sorbus torminalis, Carpinus sp, Cedrela heliconia, Quercus petraea, Zelkova sp, Carya alba, Ampelopsis cordataeformis, Berberis sp, Rhamnus saxatilis, Acer platanoides, Celtis sp, Hedera helix, Malus sp, Corylus sp, Liriodendron procaccinii, Populus aff populina, Acer opulifolium, Alnus glutinosa, Crataegus oxyacantha, Persea brauni, Philadelphus sp, Crataegus sp, Populus cf catalpa, Potamogeto sp, Sorbus sp, Alnus viridis, Paulownia catalpa, Physocarpus opulifolius, Potamogeton crispus, Pterocarya sp, Tilia alba, Acer italum, Buxus sempervirens, Cercidiohyllum crenatum, Hedera sp, Nyssa sp, Quercus alba, Quercus cerris, Quercus pretaera iberica, Aristolochia pliocenica, Celtis trachytica, Ceratophyllum, Fraxinus pliocenica, Phellodendron sp, Quercus schweitzerii, Rosa sp, Sassafras sp, Betula luminifera, Cercidiphyllum sp, Juglans acuminata, Najas marina, Pistacia sp, Pyracantha coccinea, Quercus cedrorum, Sorrbus grabbrensis, Ulmus longifolia, Acer decipiens, Aesculus pavia, Carya porcina, Leguminosites sp1, Potamogeton acutifolius, Potamogeton perfoliatus, Rhus typhina, Ribes sp, Torreya nucifera, Aesculus sp1, Aesculus sp2, Aesculus sp3, Alnus angustissima, Alnus incana, Alnus sp4, Ampelopsis sp, Buxus sp, Caranaga sp1, cf Magnolia sp, Comptonia acutiloba, Cotinus aff coccygea, Diospyros sp, Epimedium sp, Equiselum schmidtii, Fagus cf orientalis, Fraxinus exceloior, Glyptostrobus europaeus, Hedera multinervig, Kerria sp, Koelreuteria sp, Populus seinensis, Potamogeton comprensis, Potamogeton densus, Potamogeton pertina, Prunus spinosa, Pterocarya fraxinifolia, Pyrus sp, Quercus cf libani, Quercus polycarpa, Quercus sativa, Quercus sessiliflora, Salicaceae, Salix caprea, Salix cf grandifolia, Salix cinera, Salix sp1, Salix sp2, Salix sp3, Salix sp4, Salix sp5, Sambucus sp, Scirpus sp, Sorbus ariaefolia, Sorbus cf forminalis, Sorbus grandifolia, Symplocos sp, Tilia argentea, Tilia cf americana, Tripterygium sp, Ulmus scabra, Vitex agnus castus, Ziziphus sp.

In Berga (Germany, late Pliocene) the non-statistical species are: Ulmus cf carpinoides, Quercus pseudocastanea, Ulmus pyramidalis, Acer sp, Hedera helix, Pterocarya paradisiaca, Fagus sp, Platanus cf platanifolia, Quercus roburoides, Salix varians, Ulmus sp, Liquidambar europea, Parrotia persica, Populus tremula, Vitis sp, Aesculus cf hippocastanum, Acer subcampestre, Actinidia sp, Aesculus sp, Celtis sp, Juglans sp, Osmunda heeri, Quercus castaneifolia, Salix sp, Sassafras ferretianum.

In Bernasso (Germany, late Pliocene) the non-statistical species are (from Leroy & Roiron, 1996): Celtis sp, Tilia sp, Acer opalus, Acer opulifolium, Vitis sp, Populus tremula, Prunus sp, Acer sp, Acer integerrinum, Ilex sp, Hedera helix, Acer potojaponicum.

Click here for additional data file.

Supplemental Information 4 Data S1. Significance of the axes 1 and 2 from the PCAs presented in Fig. 6.

Click here for additional data file.

Supplemental Information 5 Raw data for Berga.

This contains the bulk data used for the present study.

Click here for additional data file.

Supplemental Information 6 Raw data for Bernasso.

This contains the bulk data used for the present study.

Click here for additional data file.

Supplemental Information 7 Raw data for Willershausen.

This contains the bulk data used for the present study.

Click here for additional data file.

Supplemental Information 8 PCA data for Berga.

Click here for additional data file.

Supplemental Information 9 PCA data for Bernasso.

Click here for additional data file.

Supplemental Information 10 PCA data for Willershausen.

Click here for additional data file.

We wish to thank Tony Jijina (University of Wyoming, Laramie, WY, USA), Gentry Catlett (Miami University, Oxford, OH, USA) and Samantha Moody (Bonn University, Germany) for English improvement in our manuscript. Thank you to Dr. Alexander Gehler from Geoscience Centre of the University of Göttingen for giving us access to Willershausen’s fossil collections. Special thanks to Dr. Allowen Evin (ISEM - Montpellier) and Dr. Zdeněk Janovský (Charles University, Prague) for their advices on statistical methods. We are grateful to the anonymous reviewers and Dr. Kenneth De Baets who significantly contributed to improve our manuscript. This article is the ISEM contribution n° ISEM 2018-102.

Additional Information and Declarations

Competing Interests

Author Contributions

Data Availability

The authors declare that they have no competing interests.

Benjamin Adroit conceived and designed the experiments, performed the experiments, analyzed the data, contributed reagents/materials/analysis tools, prepared figures and/or tables, approved the final draft.

Vincent Girard conceived and designed the experiments, analyzed the data, contributed reagents/materials/analysis tools, authored or reviewed drafts of the paper, approved the final draft.

Lutz Kunzmann contributed reagents/materials/analysis tools, authored or reviewed drafts of the paper, detailed on Berga location.

Jean-Frédéric Terral contributed reagents/materials/analysis tools, authored or reviewed drafts of the paper, approved the final draft.

Torsten Wappler conceived and designed the experiments, analyzed the data, contributed reagents/materials/analysis tools, prepared figures and/or tables, authored or reviewed drafts of the paper, approved the final draft.

The following information was supplied regarding data availability:

The raw data are provided as Supplemental Files.

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
