# Peer review of "Plant-insect interactions patterns in three European paleoforests of the late-Neogene—early-Quaternary"

_PeerJ, doi:10.7717/peerj.5075_

## Round 0.1 · original submission · Major Revisions

You provide an incredible amount of sound data which will be of interest for generation to come as well as an interesting new idea making this an interesting and worth published. However, I just like the reviewers feel there are still some points to be addressed before publication:

Introduction/Abstract: A clearer focus in the introduction on what you will and can do with your material would be helpful. This would include defining clearly your aims and hypotheses (see comments by reviewers 1 and 2). The abstract should also be more in line of your study – the first impression is that the focus of your paper is the study of plant-insect interaction although your focus lies on understanding the differences between different sites and their relationship with climatic factors.

Figures of types of damages: it might be helpful to illustrate examples of each type of damage (see comments by reviewer 2). It would suggest one from each site or at least one in good and one in poorer preservation. I am aware this is based on previous work, but it would be easier to follow for readers who don’t have access to the literature or particular journals. Potentially, they could also be added to figure 6 (if it doesn´t become too crowded).

Ordination: Not sure if a PCA is the best method for your purpose (see comment by reviewer 2) – NMDS might also be useful. You at least need to specify better why and how you used the method.
Potential relationship with climate or seasonality: reviewer 2 found the interpretation of the potential relationship with seasonality quite speculative. I feel this is still important and worth discussing but clearly need to separate what can be said based on the available data and what needs to be further tested in the feature. You provide a wealth of important data on 3 paleoforests, but probably more is necessary to disentangle the relative effects of various drivers. You could highlight this in the conclusion that this is just a first step in that direction – that would be fine (and in line with PeerJ guidelines)

Correlation between climate parameters and plant diversity: Be careful here. You did not do any statistical tests on this making it hard to be sure. Furthermore, with data from just 3 sites this would be really hard to do anyway. I suggest keep this for the discussion (and highlight it as a possible hypothesis to be further tested) or you focus more on other aspects of the manuscript.

Language: There were some odd phrases or expression in the text so that it might be worth to let a colleague native or fluent in English read your article before resubmission.

Please in addition to the points raised by the reviewers, also address the following points:
Line 102: “tumbled”: please rephrase – this sounds odd (do you mean transported?)
Line 107: please add “initially described” or “extensively described” in from of “by (Von Koenen, 1895)
Line 122: it would be helpful to add the common name(s) for Parrotia persica and Liquidambar europaeum to make it easier to follow for non-experts.
Line 147: “shut” sounds odd in this context; do you mean “closed off” or “restricted”?
Line 161: CLAMP – I am aware what CLAMP analysis is, but other readers might not; please spell out the abbreviation for completeness sake.
Line 183: new values of what? Please specific for general readability
Line 230: it would refer to “poorer preservation” rather than bad preservation as it means the same, but less negative and more correct in this context
Line 306: Could there also be “more time” (e.g., large temporal interval) in the Willershausen outcrop which might blur things
Line 328: Please reformat to Thiel, Klotz & Uhl (2012)
Line 345: “through” instead of “trough”
Line 360; “linked” instead of “link”
Line 363: please reformat to Leroy & Roiron (1996)
Line 409: “Girard et al., in review” can only be cited once published or at least accepted for publication or made online available as preprint; otherwise, rephrase as “(Girard et al., unpublished)”
Line 420; rephrase to “that a negative correlation exist between …”
Figure 3: some type of errors bars would be helpful and meaningful – binomial confidence intervals or standard errors (see Möller et al. 2017)

Reviewer 1 ·

Basic reporting

Your paper will be pretty important for the paleobotany and plant-insect interactions field of study. It's clear, with good data and relevant results.

Experimental design

You need be more specific about what is your main goal and hypothesis. You have really good data. By your introduction the reader gets the idea that you will just describe the damages, in a qualitative way.
Also, you can improve the description of your analysis.

Validity of the findings

You presented strong results and an interesting discussion. I think you can improve your conclusions when you specify your hypothesis in the beginning.

Additional comments

Despite of the lack of a clear question, I believe that your paper will provide a significant contribution to the study of fossil plant–insect interactions. See my suggestions in the attached document.

Annotated reviews are not available for download in order to protect the identity of reviewers who chose to remain anonymous.

Reviewer 2 ·

Basic reporting

Please see attached pdf with all comments.

Experimental design

Please see attached pdf with all comments.

Validity of the findings

Please see attached pdf with all comments.

Additional comments

Please see attached pdf with all comments.

Annotated reviews are not available for download in order to protect the identity of reviewers who chose to remain anonymous.

---

## Round 0.2 · Minor Revisions

Thank you for addressing our suggestions. Your manuscript is as good as accepted, there is just some minor points I would like you to address before publication. These are mostly related to the way things are formulated (see annotated pdf), but some points are more important:

Lines on human influence in the abstract: The non-disturbed ecosystems in the geological past are an important point to make, but it think it can be done in a more concise way in the abstract (as it is not your focus). I would not cut all the lines (as suggested by the reviewer), but the point you want to make can be done in half or even one third of the amount of lines you currently dedicate to it (see comment).

Effect of biodiversity and sample size: I agree that the diversity of leafs in museum collection(s) does not necessarily reflect the diversity of the forest. You do however state that the pattern is the same for common taxa as for the whole. The biodiversity of the available leaves still might have a major impact on the measured prevalence of damage categories. You analysed the statistical significance of differences of whole community or only more common taxa, but I still feel it might be useful to provide a graph in the supplementary showing the standard error on these percentages. Particularly for some species or genera which are (more commonly) distributed in all samples and show interesting patterns. As for the effect of the preservation, I suspect the disappearance of damaged versus undamaged leaves should be similar among all studied sites – I at least cannot imagine how a straightforward bias in one or the another direction could be caused using differential taphonomy, but I am not a paleobotanist nor a paleoentomologist. I agree with you that the difference between Bernasso and the German sites is probably an environmental one. In this respect it is interesting to highlight more that Bernasso has lower diversity, but more DT´s. This is mentioned in Lines 230-232, but could be highlighted more in the discussion (e.g., for further support that biodiversity/sampling isn´t everything).

Figures: There are still some issues with the figures and their captions (see also comments by reviewer). Please make sure the y-axes are correctly labeled in Figure 4. Figure 5 is easier to follow, but would it might be worthwhile to consider add convex hulls around some genera or species common to two or more sites which you discuss in more detail in lines 271-286 (e.g., Acer, Carpinus, Carya, Parrotia, Sorbus, Zelkova).

As for the statistical analysis; I agree with your arguments that PCA is probably best for your dataset now that the rationale of the analysis is more extensively described. For future reference, I wanted to point out that there are methods to also make more meaningful comparisons of distances between samples also with NMDS (e.g., some of the approaches implemented in Vegan implemented in R which repeat the analyses and rescale the axes).

Please consider comments in the annotated pdf and points raised by the reviewer, in addition to these points.

Reviewer 1 ·

Basic reporting

I think your manuscript really improved since last review. It's much more clear and direct to the point. I have just some minor suggestions.

Experimental design

Your goal and hypothesis are much clear now. And your methodology properly helps to answer your questions.

Validity of the findings

You have really interesting results. I just have some suggestions to improve your discussion (see document attached)

Additional comments

Despite of my suggestions, I believe that your paper will provide a significant contribution to the study of fossil plant–insect interactions.

Annotated reviews are not available for download in order to protect the identity of reviewers who chose to remain anonymous.

---

## Round 0.3 · accepted · Accept

Thank you for making this final changes and updating the figures. I feel the manuscript is even easier to follow. Looking forward to seeing this published.

#